# Toxic Cyanopeptides Monitoring in Thermal Spring Water by Capillary Electrophoresis Tandem Mass Spectrometry

**DOI:** 10.3390/toxins17020063

**Published:** 2025-01-31

**Authors:** Rocío Carmona-Molero, Laura Carbonell-Rozas, Ana M. García-Campaña, Monsalud del Olmo-Iruela, Francisco J. Lara

**Affiliations:** Department Analytical Chemistry, University of Granada, Av. Fuente Nueva s/n, 18071 Granada, Spain; rociocarmona@ugr.es (R.C.-M.); rozas@ugr.es (L.C.-R.); amgarcia@ugr.es (A.M.G.-C.); mdolmo@ugr.es (M.d.O.-I.)

**Keywords:** capillary electrophoresis, green analytical chemistry, mass spectrometry, salting-out liquid–liquid extraction, thermal spring water, toxic cyanopeptide

## Abstract

Cyanobacteria are an ancient group of prokaryotes capable of oxygenic photosynthesis. Recently, thermal crises symptoms in hot springs have been associated with acute cyanopeptides poisoning. The aim of this work is to develop a fast, easy and reliable method to monitor the presence of toxic cyanopeptides in geothermal waters. The analytical method based on capillary zone electrophoresis coupled with tandem mass spectrometry (CZE-MS/MS) was developed for the simultaneous determination of 14 cyanopeptides in less than 7.5 min. A basic 50 mM ammonium acetate buffer at pH 10.2 was selected as the background electrolyte, positive electrospray ionization (ESI+) was employed for all compounds, and a salting-out assisted liquid–liquid extraction (SALLE) protocol with acetonitrile as an extraction solvent and MgSO_4_ as an auxiliary salting-out agent was optimized as sample treatment. Six natural hot springs in the province of Granada (Andalucía, Spain) were sampled at the beginning of the summer season (June) and at the end (September). Biomass collected at two sample points (Santa Fe and Zújar) contained cyanobacteria cells from the genera *Phormidium*, *Leptolyngbya,* and *Spirulina*. Nevertheless, cyanotoxins covered by this work were not found in any of the water samples analyzed. The greenness and transferability of the method was evaluated highlighting its sustainability and applicability.

## 1. Introduction

Cyanobacteria or blue-green algae are photosynthetic prokaryotes producing potentially harmful compounds in the form of secondary metabolites called cyanotoxins [1], being one of the most important groups of natural toxins [2]. Eutrophication and other processes, such as global warming, have been shown to be responsible for cyanobacterial harmful algal blooms (CyanoHABS), i.e., an uncontrolled increase in time and space of these organisms. Thus, cyanobacteria reproduce exponentially, and their presence extends to all aquatic and terrestrial ecosystems [3]. They have been reported as well in polar regions, deserts, hypersaline environments, alkaline lakes, and hot springs [4]. CyanoHABs can be easily identified as they modify water properties such as turbidity, odor, or taste [5]. In addition, it has a negative impact on water quality, ecosystem integrity, and human health [6]. More than 40 genera of cyanobacteria are toxin producers and up to 10 different cyanotoxins groups—which, in turn, have multiple variants—have been studied to date [7]. However, only one variant of cyanotoxin has been regulated in tap water (1 μg/L for microcystin-leucine-arginine (MCLR)) [8].

Due to the vast number of members, most studies based on the detection of cyanotoxins include cyclic peptides such as microcystins (MCs) and nodularins (NODs) [9,10]. Among several NOD variants, Nodularin-R is the most studied. NOD is synthesized by the cyanobacteria *Nodularia spumigena*, which is found in brackish waters [10]. MCs are produced by different cyanobacteria genera, such as *Microcystis*, *Anabaena*, *Plankthotrix*, *Aphanizomenon*, *Anabaenopsis*, *Nostoc*, *Rivularia,* and *Fisherella* [11] that occupy all water systems, from brackish to marine and freshwater. They are normally inside cells until cyanobacterial lysis. More than 279 variants of this heat-stable and water-soluble cyclic peptide have been identified [12]. Its toxicity extends to humans, domestic animals, livestock, aquatic organisms, plants, and eukaryotic microbes because they are potent inhibitors of serine/threonine protein 1 (PP1) and 2A (PP2A) phosphatases as well as disruptors of intracellular homeostasis [13,14,15]. The inhibition of these phosphatases disarranges the cell cytoskeleton, causes cell death, leads to intrahepatic hemorrhage, and sometimes kills exposed organisms. MCs have been categorized as possible carcinogens by the International Agency for Research on Cancer (IARC), as they can promote liver tumors [16]. Within the group of cyclic peptides, anabaenopeptins (AP) have been less investigated. They are produced by the genera *Anabaena*, *Planktothrix*, *Nodularia*, and *Microcystis* and include over 96 congeners [17]. APs often occur with microcystins [18], and even though they may occur more frequently than MCs, they have not received attention in terms of potential human or ecological effects [19]. Therefore, its toxicity to humans and other mammals is still unknown.

Most scientific studies focus on cyanotoxins in temperate freshwater ecosystems and some coastal areas. However, the literature on extreme ecosystems, such as hot springs, is still scarce despite their interest due to the unique thermophilic properties of the organisms that thrive in these niches. Temperature changes, pH, concentration of free sulfide, and metabolic requirements determine the distribution of thermophilic cyanobacteria [4]. MC-containing mats have been identified in hot springs characterized not just by high temperature, but also by elevated values of pH, conductivity, and moderate salinity [20]. Predominant cyanobacterial mats in those environments include some species of the *Oscillatoria*, *Phormidium*, *Spirulina*, and *Synechococcus* genera producing MCLR, microcystin-arginine-arginine (MCRR), microcystin-leucine-phenilalanine (MCLF), and microcystin-tyrosine-arginine (MCYR). Filamentous stigonematalean cyanobacteria are also known to be present in geothermal habitats [4]. In relation to this, the “thermal crisis” concept has recently emerged from the need to englobe several digestive, hepatic, and cutaneous discomforts after recreational use of thermal water sources. This clinical feature has been directly related to the presence of cyanotoxins in the water bodies, with MCs being the most frequently reported cyanotoxins in these extreme environments [21].

In recent years, capillary electrophoresis coupled with tandem mass spectrometry detection (CE-MS/MS) methods has become an attractive alternative in food, environmental, pharmaceutical, as well as in omics analysis, providing not only high-resolution separations, but also unequivocal identification, enabling molecular characterization based on fragmentation [22,23,24]. There has been a growing interest in the development of sustainable analytical methodologies in compliance with green analytical chemistry (GAC) principles. In this regard, several metric tools have been recently developed to assess the greenness of the analytical methods [25]. Among them, AGREEE (Analytical GREEness) is an open access software based on the 12 principles of GAC (significance) [26], which have been transformed into a unified 0–1 scale and presented with a final score in a pictogram [27]. In addition, another metric to evaluate the practicability of the analytical method, the blue applicability grade index (BAGI), was recently proposed [28]. BAGI can be considered complementary to the well-established green metrics, and it is mainly focused on the practical aspects of White Analytical Chemistry (WAC) [29]. This metric considers 10 criteria to produce a pictogram and a score that depicts the applicability and functionality of an analytical method. Among the demanded analytical techniques which prevent waste production, use safer solvents, and improve energy efficiency, CE is an analytical technique considered sustainable and green as it uses inexpensive fused silica capillaries, low sample size, and reagent volumes, and produces reduced chemical waste. In the literature, CE is suggested to replace common LC methods as a feasible alternative in terms of results and the greener distinction [30]. Reversed-phase liquid chromatography coupled with mass spectrometry (LC-MS) has been the technique of choice for cyanotoxin determination, mainly for those analytes belonging to the same family [31,32]. To achieve multi-class cyanotoxin separation, CE-MS/MS [33] has been shown as a competent option against hydrophilic interaction liquid chromatography coupled with tandem MS (HILIC-MS/MS) [34], due to its cheaper cost, low reagent consumption, and shorter re-equilibration steps.

Solid-phase extraction (SPE) is the most common sample treatment for cyanotoxins, in both off-line [35,36] and on-line [37] configurations. Nevertheless, lower solvent consumption and less time-consuming procedures are major aims in recent sample clean-up/enrichment handling. Salting-out assisted liquid–liquid extraction (SALLE) is simpler, fast, uses non-pollutant salts, and requires a small volume of organic solvents which are easily dried and reconstituted in a suitable solvent for CE separation. This methodology has been previously reported for the extraction of microcystins prior to their analysis by the LC-MS/MS method [38].

In this study, we propose for the first time a capillary zone electrophoresis coupled with the tandem mass spectrometry (CZE-MS/MS) method to analyze 14 cyclic cyanopeptides: MCLR, MCRR, MCLF, MCYR, microcystin-leucine-tyrosine (MCLY), microcystin-homotyrosine-arginine (MCHtyR), D-aspartate-microcystin-leucine-arginine ([D-Asp3] MCLR), microcystin-isoleucine-arginine (MCHilR), microcystin-leucine-alanine (MCLA), microcystin-tryptophan-arginine (MCWR), microcystin-leucine-tryptophan (MCLW), anabaenopeptin A (AP A), anabaenopeptin B (AP B), and NOD in various thermal water sources from Granada (Spain), obtaining the lowest detection limits reported in both LC and CE so far to the best of our knowledge.

## 2. Results and Discussion

### 2.1. Optimization of the Electrophoretic Separation

The 14 analytes included in this work are large cyclic peptides whose molecular masses range from 825.0 to 1068.3 g/mol (Appendix A). Among them, the 11 MCs, ranging from 910.1 to 1068.3 g/mol, converge in a cyclic heptapeptide sharing a constant five conserved amino acids and two variable L-aminoacids at positions 2 and 4 (cyclo-(D-Ala^1^)-X^2^-(D-MeAsp^3^)-Z^4^-Adda^5^-(D-Glu^6^)-Mdha^7^), which mainly confer the diversity of MC congeners along with demethylations [39]. The exact substituent of each MC is found in Appendix A. Therefore, every MC has a similar fragmentation pattern, and all of them present the Adda fragment characteristic for this group of cyanotoxins. The β-amino acid Adda has not been reported elsewhere in nature apart from the closely related group of NODs [40].

The electrophoretic optimization was carried out in Selected Ion Monitoring (SIM) mode. Monoprotonated and di-protonated molecular ions were used throughout the optimization process (Appendix A).

Firstly, a background electrolyte (BGE) composition was set. Micellar electrokinetic capillary chromatography using ammonium perfluorooctanoate (APFO) as an anionic surfactant in the BGE was thought to be the best option due to the hydrophobicity of most analytes [41]. However, results were not satisfactory in terms of signal intensity and other separation modes were tested. Wide pH (2.1–10.5) and BGE compositions (APFO, formic acid (FA) in water, ammonium acetate, ammonium carbonate, and ammonium formate) were tested. Ammonium acetate acidified with FA provided the best peak efficiencies, but a basic ammonium acetate buffer showed higher peak heights for all analytes, thus, further investigation regarding the pH of the ammonium acetate buffer was needed. The effect of pH around ammonia (NH_3_) pKa (9.25) in 40 mM ammonium acetate was then studied. An increase in peak efficiency (plates per meter) was noticed at pH 10.2. Afterwards, ammonium acetate concentration was evaluated. Different concentrations, from 20 to 50 mM, were tested. A concentration of 40 mM ammonium acetate showed the best peak heights; however, tailed peaks were observed at this concentration. The addition of organic modifiers (isopropanol (IPA), ethanol (EtOH), methanol (MeOH), and acetonitrile (MeCN)) that are normally used to improve separation process was also investigated. Any of these modifiers improved peak efficiency at this concentration of ammonium acetate, increasing also the analysis time. The tailed peaks disappeared when using 50 mM ammonium acetate and 186 mM NH_3_ (pH 10.2) as BGE, showing the best peak efficiencies and an acceptable analyte signal. The influence of the ammonium acetate concentration can be observed in Figure 1, where the extracted ion electropherograms (EIEs) at 40 and 50 mM ammonium acetate are compared.

The length of the capillary was evaluated from 70 to 90 cm. Capillary length was set at 80 cm since no improvements were observed when increasing the length to 90 cm, and undesired electrophoretic current increase was noticed when shorter 70 cm capillaries where employed. Maximum voltage allowed by the software (30 kV) provided the best sensitivity for the studied analytes, therefore it was selected. The temperature of the cassette was studied from 15 to 25 °C. An improvement of some analyte signals was noticed at 25 °C, so it was selected for further experiments. Optimum conditions result in a baseline resolution of the 14 cyanotoxins in less than 7.5 min, with an electric current during the separation of 41 µA.

### 2.2. Optimization of Detection Parameters

Electrospray in positive mode (ESI+) was used in this work. Different parameters affecting the electrospray ionization source, such as sheath–liquid composition and its flow rate, dry gas flow rate and temperature, and nebulizer pressure have been optimized considering the signal-to-noise ratio (S/N) as the response variable. Sheath–liquid composition should be carefully selected in CE-MS/MS methods in order to ensure the stability of the electrospray and a good sensitivity. Thus, MeOH, MeCN, IPA, and EtOH were evaluated as organic solvents, and FA and acetic acid (AA) as acids. MeOH showed a clear improvement in the S/N of all analytes. Its percentage was fixed at 30% as it favored the signal (peak height) for most of the analytes (Figure 2).

For acid content, 0.3% FA was established as better S/N was achieved for all compounds in the tested range (from 0.1 to 0.4% FA). Sheath–liquid flow rate was studied from 2 to 20 µL/min, and 10 µL/min was selected to obtain a stable electrospray. The rest of the ESI+ parameters were optimized as follows: sheath gas temperature (130 to 250 °C) was set at 195 °C; sheath gas flow rate (3.5 to 8 L/min) was set at 5 L/min; dry gas temperature (80 to 200 °C) was set at 150 °C; dry gas flow (11 to 15 L/min) was set at 11 L/min as this was the minimum allowed by the software; nebulizer pressure (6 to 14 psi) was established at 12 psi; and for capillary and nozzle voltage ( both 1000 to 3000 V) 2000 V were selected.

MS/MS transitions were then studied. Firstly, SCAN mode was used to select the precursor ions. Di-protonated molecular ions were observed for most analytes. A similar fragmentation pattern was previously observed for the cyclic peptides analyzed by this technique [33]. Once the precursor ion was confirmed, SIM mode was used together with the previously optimized conditions. The most intense fragments were then selected to optimize multiple reaction monitoring (MRM). For each transition, optimum collision energy was studied to increase the signal as much as possible. Dwell time was also evaluated for each case to ensure data acquisition with a minimum of 10 points per peak (Appendix A). Mono-protonated AP B, Na^+^ adduct of di-protonated MCWR, and di-protonated MCLY present in the SIM method (Appendix A) were discarded as precursor ions as higher signals were found with other precursor ions. Three time-segments have been established, from 0.10 to 5.40, 5.40 to 5.95, and 5.95 to 7.50 (Appendix A). This allowed for higher dwell times and therefore significantly improved sensitivity.

### 2.3. Optimization of the On-Line Preconcentration Strategy

On-line sample preconcentration is an effective approach where the analytes are focused into a constricted band after sample injection, and therefore, method sensitivity is increased. Among existing methodologies, pH-mediated stacking and field-amplified sample stacking (FASS) could be considered as the simplest and most used techniques [42]. FASS is based on the difference in conductivity between the BGE and the injection solvent, leading to different electrophoretic mobilities of the analytes in each medium.

Initially, different preconcentration strategies were tested: FASS, field-amplified sample injection (FASI), pH-junction, and barrage pH-junction. At neutral pH (7), most analytes are negatively charged, except for MCRR that has predominantly neutral charge. When voltage is applied, analytes would move back to the anode because of their negative charge, and the electroosmotic flow will push them to the cathode. In the interphase between the injection solvent and the BGE, conductivity is higher, and analytes will slow down and stack up. This preconcentration effect was even more intense when the sample was dissolved in 200 mM NH_3_, since the number of negative charges increases. However, this sample solvent was discarded because it caused frequent current disruptions. Regardless of this, injection in ultrapure water led to satisfactory preconcentration results, although, a small percentage of MeOH was added to maintain analytes dissolved given their polarity. A final injection solvent composed of H_2_O:MeOH (95.5:4.5, *v*/*v*) achieved enrichment factors (EFs) of up to 5-fold for the analyte signals, and significantly improved peak efficiencies (Figure 3 and Figure 4). EFs were calculated as the ratio between peak heights when stacking was applied and peak heights when stacking was not applied (sample solvent: BGE). Injection time was increased until peak efficiency decreased (50 s).

### 2.4. Salting-Out Liquid–Liquid Extraction (SALLE) Protocol

The optimization of SALLE as sample treatment was carried out using a natural thermal water sample from Santa Fe (Granada, Spain) as a representative matrix. Firstly, salt nature was studied. Sodium chloride (NaCl), ammonium sulfate ((NH_4_)_2_SO_4_), magnesium sulfate (MgSO_4_), and sodium sulfate (Na_2_SO_4_) were tested at a given amount (1.6 g). In general, MgSO_4_, (NH_4_)_2_SO_4_ and Na_2_SO_4_ provided good recoveries, but Na_2_SO_4_ was discarded because it worsens peak efficiency. MgSO_4_ was the only salt ensuring recoveries near or superior to 80% for all analytes, especially MCRR (Figure 5).

The organic solvent used in this extraction was limited to MeCN. EtOH and IPA were discarded as phase separation was not achieved even by increasing the amount of the salting-out agent. MeOH and tetrahydrofuran (THF) had a polarity lower than necessary to extract our analytes.

The salt amount was studied from 1.8 to 3.6 g. amounts. Phase separation was not achieved with lower amounts. Both 1.8 and 2.4 g provided high recoveries, but with 2.4 g, the extracts were dried faster under the nitrogen flow and peak efficiency was notably better. MeCN volume was also studied from 1 to 2.5 mL and set at 2 mL as this gave satisfactory recoveries (above 80% in all cases). Finally, sample volume was tested from 5 to 10 mL, 7.5 mL was selected as recoveries, and matrix effects were similar as for 5 mL, but the preconcentration was higher. The amount of 10 mL was discarded due to the increase on the matrix effect. Centrifuge time was set at 5 min with 5000 revolutions per minute (rpm) since it was the minimum time to adequately separate the organic phase and the maximum rpm that glass tubes support. Agitation time was set at 15 min to obtain reproducible performance and satisfactory results.

### 2.5. Method Characterization in Thermal Spring Water

The SALLE-CZE-MS/MS methodology developed was applied to thermal spring water samples from the province of Granada (Spain): Alicún de las Torres, La Malahá, Alhama de Granada, Urquízar (Dúrcal), Zújar and Santa Fe. Conductivity, pH, and temperature in the water samples are summarized in Appendix A. The method was characterized in terms of linearity, precision, limits of detection (LODs), quantification (LOQs), recoveries, and matrix effects.

#### 2.5.1. Calibration Curves and Analytical Performance Characteristics of the Method

Procedural calibration curves were performed in Santa Fe thermal water. Procedural calibration involves the analysis of samples spiked before the sample treatment. This calibration, which compensates not only the matrix effect, but also the sample treatment losses, ensures the reliability of the quantification. Three samples per concentration level were prepared and analyzed in triplicate (*n* = 9) according to the SALLE-CZE-MS/MS proposed method. The peak area, demonstrated as signal response versus analyte concentrations, was monitored. A blank sample was also analyzed, and no interfering signals were detected at the migration time of the analytes (Figure 6). LODs and LOQs were obtained as the lowest concentration producing a signal-to-noise ratio of three and ten, respectively (Table 1).

The coefficient of determination (R^2^) is above 0.992 in all cases. LOQs are below the MCLR tap water regulation of 1 µg/L. LOQ and LOD are significantly lower than previously published methods based on both CE-MS [43,44] and LC-MS [32,38,45] for determining microcystins and nodularin in water samples.

#### 2.5.2. Matrix Effect, Recoveries, and Precision Assays

The matrix effect (ME) was calculated as the relationship between the peak area of a blank sample spiked after the extraction procedure and the peak area of a standard solution at the same analyte concentrations (Table 2). Matrix effects were lower than|12|% in all cases.

Recoveries (R, %) for the 14 cyanopeptides were calculated to evaluate the extraction efficiency of the sample treatment (SALLE) as the ratio between the peak area of samples spiked before the sample treatment and the peak area of the samples spiked after sample treatment. Both ME and R were calculated considering three experimental replicates injected in triplicate in the CE-MS/MS system (*n* = 9) for each level. We can see in Table 2 recoveries of all analytes above 85%. ME and R demonstrate that the use of MgSO_4_ instead of (NH_4_)_2_SO_4_ and a detailed optimization of sample volume provides better results than previously published methods [38].

The precision of the method was measured in terms of repeatability (intra-day precision) and intermediate precision (inter-day precision). For repeatability, during the same day and identical conditions, three samples for each concentration level were analyzed in triplicate (*n* = 9). For intermediate precision, during five consecutive days, five samples were analyzed in triplicate (*n* = 15).

The results of precision assays, expressed as relative standard deviation (RSD, %) of the peak areas, showed adequate intra-day and inter-day values below 15% in both cases. These results are slightly better than those obtained from LC methods [34,38,45,46].

To confirm the applicability of the method in the rest of the thermal water samples, ME and recoveries were calculated for Zújar (Appendix A) and Dúrcal (Appendix A) sampling points. In both samples, recoveries were above 80% and ME below |20|%, except for AP B in Dúrcal, where a 62.04% ME was found in L2. Of the three water samples analyzed, Dúrcal is the one that differs the most from the others, as conductivity is lower, and temperature is more than 10 °C below Zújar and Santa Fe (Appendix A); these could be related with the slight differences in the results obtained.

### 2.6. Analysis of Thermal Spring Water Samples

Natural hot springs of the province of Granada were sampled at two times of the year; first, at the beginning of the summer season (June) and second, at the end of the high heat season (September), trying to match the second sampling after cyanobacterial cell lysis. Sampling points can be found in Appendix A. Analysis performed using the optimized CZE-MS/MS method show that no target compounds were identified at any of these sampling points. In the June sampling, biomass was only found at three of the six locations: Dúrcal, Zújar, and Santa Fe. All the other spots were discovered to be previously filtered and, in some cases, chemically treated by private thermal resorts, so they were discarded for the study because they are outside the scope of this work. Biomass was analyzed and *Spyrogyra* microalgae was found in Dúrcal, having a morphology easily mistaken with cyanobacteria *Spirulina* (Figure 7). Cyanobacterial mats were found at Zújar and Santa Fe. Three different cyanobacteria genera were found at Zújar, classified according to their abundance: *Phormidium*, *Leptolyngbya,* and *Spirulina*. *Phormidium* and *Leptolyngbya* were also present in Santa Fe, with the second one being the most abundant. The *Phormidium* genus is known to produce cyanotoxins and some species of *Leptolyngbya* have been shown to produce MCs and NOD as well [47].

A previous work on thermal spring water did find positives in biomass samples, but not in water samples [21]. We found similar results when we tried a tentative determination of cyanotoxins in biomass samples from Santa Fe and Zújar; AP B was found above the detection limit in Zújar’s biomass samples.

### 2.7. Greenness and Practicability Evaluation

Considering the importance of aligning with new trends in GAC, we have evaluated the proposed SALLE-CZE-MS/MS method by using the AGREE and BAGI tools. In addition, the resulting scores were compared to those of a previously reported SALLE-UHPLC-MS/MS method for determining cyanotoxins in water samples [38]. This work was selected to ensure a fair comparison of the methodologies since both target compounds and sample of interest were similar.

The main characteristics of these methods, as well as the AGREE and BAGI scores, are shown in Table 3.

Regarding the greenness assessment by AGREE, it can be observed that criterion 2, regarding the sample size, was slightly better in the case of SALLE-UHPLC-MS/MS since a smaller volume of water sample was used. Although minimal sample size is recommended, in the proposed method, the sample volume was increased to guarantee the preconcentration of the target compounds, and therefore, to increase sensitivity of the method. In terms of waste generation (criterion 7) and the use of toxic reagents (criterion 11), both are higher in LC due to the use of organic mobile phases, particularly when compared to the BGE used in CE. Consequently, both criteria were improved with the proposed CZE method. As can be observed in Table 3, the proposed method involved the determination of a higher number of compounds in a shorter total analysis time. The sample throughput (samples analyzed per hour) in multi-analyte methods is preferred; that way, criterion 8 is slightly greener for the CZE-MS/MS method. Although the score could be improved, this comparison highlights that the proposed CE method can be considered as a greener alternative to LC methods.

In relation to the applicability of the methods, in both cases, the BAGI score was above 60 points as it is recommended by the authors to consider an analytical method as “practical”. In fact, the improvement of the BAGI score when using CZE-MS/MS was mainly due to the higher number of samples that can be analyzed per hour (attribute 6) and the number of compounds (attribute 2). This evaluation highlighted that the proposed method could be easily applied in other laboratories equipped with CE technology. Nevertheless, this methodology is still not widely used as LC in routine or reference laboratories. In this framework, the demonstration of the greenness and applicability of the CZE-MS/MS method is crucial to promoting the use of this technology in food safety, considering its advantages in terms of sustainability as well as its practicality.

## 3. Conclusions

A CE-MS/MS method for the simultaneous determination of fourteen cyclic peptides cyanotoxins: eleven microcystins (MCLR, MCHilR, MCHtyR, [D-Asp3], MCLR, MCLF, MCRR, MCLA, MCLW, MCWR, MCYR, MCLY), two anabaenopeptins (AP A and AP B) and one nodularin (NOD) has been proposed. Considering the high molecular weight and the multiples pKas of each molecule, a simple FASS stacking technique has been applied for increasing sensitivity by online preconcentration. Linearity, sensitivity, recovery, and precision of the optimized method show satisfactory results in all cases.

The method has been validated in natural thermal spring water samples from the province of Granada (Spain) with better recoveries and smaller matrix effects than previous published methods with similar sample treatment. The LOQs achieved are below the levels recommended in tap water by the World Health Organization (WHO).

Although cyanobacteria producing these cyanotoxins have been identified in the biomass samples taken from two geothermal locations, cyanopeptides have not been detected in the water samples analyzed.

## 4. Materials and Methods

### 4.1. Reagents and Materials

The cyanotoxins included in this study were MCLR ≥ 99%, MCRR ≥ 99%, NOD ≥ 95%, MCLY ≥ 95%, AP B ≥ 95%, AP A ≥ 95%, MCHtyR ≥ 95%, [D-Asp3] MCLR ≥ 95%, MCHilR ≥ 95%, MCLA ≥ 95%, MCYR ≥ 95%, MCWR ≥ 95%, MCLW ≥ 95%, and MCLF ≥ 95. They were supplied by Enzo Life Sciences, Inc. (Lausen, Switzerland). Standard solutions were made by adding 1 mL of the appropriate solvent directly into the vial containing the toxin provided by the manufacturer and gently swirling the vial to dissolve the toxin. The obtained solutions were 50 μg/mL MCLR in methanol, 25 μg/mL MCRR in 50% aqueous methanol, 50 μg/mL NOD in 50% aqueous methanol, 25 μg/mL MCLY in methanol, 25 μg/mL AP B in methanol, 25 μg/mL AP A in methanol, 25 μg/mL MCHilR in methanol, 25 μg/mL MCHtyR in methanol, 25 μg/mL [D-Asp3]MCLR in methanol, 25 μg/mL MCLA in methanol, 25 μg/mL MCYR in methanol, 25 μg/mL MCWR in methanol, 25 μg/mL MCLW in methanol, and 25 μg/mL MCLF in methanol. All of them were stored in the dark at −20 °C. Intermediate standard solutions of each compound at 2.5 μg/mL were prepared by dilution of the stock solutions with the corresponding solvent for each toxin. These solutions were used to prepare the working solutions that consisted a mixture of all cyanotoxins in concentration levels according to the experiment in 50% aqueous methanol. These solutions were stored at 4 °C and equilibrated to room temperature before use.

Unless stated differently, analytical grade reagents and HPLC grade solvents were utilized in this work. Acetonitrile (MeCN), isopropanol (IPA), methanol (MeOH), and ammonium sulfate ((NH_4_)_2_SO_4_) were purchased from VWR (Radnor, PA, USA). Ethanol (EtOH), ammonia solution (NH_3_·H_2_O) (30% assay), and ammonium acetate (NH_4_CH_3_CO_2_) was purchased from Merck (Darmstadt, Germany). Formic acid (FA) and acetic acid (AA), ammonium formate (CH_5_NO_2_), and ammonium carbonate ((NH_4_)_2_CO_3_) were purchased from Sigma-Aldrich (St. Louis, MO, USA). Sodium hydroxide (NaOH), Magnesium sulfate anhydrous (MgSO_4_), sodium chloride (NaCl), and sodium sulphate (Na_2_SO_4_) were obtained from PanReac-Química (Madrid, Spain). Ultra-pure water (Milli-Q plus system, Millipore, Bedford, MA, USA) was used throughout the study. CLARIFY polytetrafluoroethylene (PTFE) hydrophilic filters (0.2 μm × 13 mm) were employed to filter the extracts.

### 4.2. Instrumentation

CE experiments were conducted using an Agilent 7100 CE system connected to a triple quadrupole 6495C mass spectrometer (Agilent Technologies, Waldbronn, Germany) that featured an electrospray ionization source functioning in positive ionization mode (ESI+). The sheath liquid was provided by a 1260 Infinity II Iso Pump. Mass spectrometry data were gathered and analyzed using MassHunter software (version 10.0).

Separations were carried out in bare fused-silica capillaries (80 cm of total length, 50 µm I.D., 375 µm O.D.) from Polymicro Technologies (Phoenix, AZ, USA).

A pH-meter with a ± 0.01 pH unit resolution (Crison model pH 2000, Barcelona, Spain), a vortex−2 Genie (Scientific Industries, Bohemia, NY, USA), an analytical balance with 0.0001 g resolution (Sartorius; Goettingen, Germany), a multi-tube vortexer (model BV1010 from Benchmark Scientific; Sayreville, NJ, USA), a Universal 320R centrifuge (Hettich Zentrifugen; Tuttlingen, Germany), and a nitrogen evaporator EVA-EC System (VLM GmbH; Bielefeld, Germany).

### 4.3. Sample Collection

Thermal water samples collected in June and September 2024 from different natural thermal spots named Zújar, La Malahá, Alicún de las Torres, Alhama de Granada, Santa Fe, and Urquízar pond (Dúrcal) located in Granada (Andalucía, Spain) were considered in this study. Biomass was also sampled if present, meaning Dúrcal, Zújar, and Santa Fe in our case. Samples were collected in amber glass bottles at different points of the corresponding thermal spot. Temperature, conductivity, and pH were measured in situ. Water samples were not filtered prior to sample treatment. Glass bottles filled with corresponding samples were kept at 4 °C until analysis.

### 4.4. SALLE

Aliquots of 7.5 mL of thermal water samples were placed in glass tubes. Afterwards, 2.4 g MgSO_4_ was added to the tube and vortexed for 5 min until its dilution. Subsequently, 2 mL MeCN was added to the tube, and it was vortexed for 10 min in a mechanical shaker. Then, glass tubes were centrifuged at 5000 rpm for 5 min. Organic phase (supernatant) was transferred to a 4 mL glass vial. The vial was placed in a nitrogen dryer and evaporated to dryness at 30 °C under a gentle stream of nitrogen. The residue was re-dissolved with 100 µL of H_2_O:MeOH (95.5:4.5, *v*/*v*). The final extract was filtered (PTFE hydrophilic filters (0.2 μm × 13 mm)), transferred to a 100 µL glass insert, and analyzed by the proposed CE-MS/MS method.

### 4.5. CE Separation

New capillaries were conditioned with 1 M NaOH for 15 min, ultrapure milli-Q water for 10 min, and BGE for 15 min at 1 bar pressure and 25 °C temperature. To achieve sufficient repeatability between runs, the capillary was rinsed with BGE for 3 min at 1 bar and 25 °C at the start of each run. At the conclusion of each working day, the capillary was cleaned using ultrapure water for 5 min and then dried with air for 10 min at 1 bar and 25 °C.

CE separation was carried out in bare fused-silica capillaries (80 cm of total length) at 25 °C. BGE consisted of 50 mM ammonium acetate 186 mM NH_3_ (pH 10.2) in ultrapure water. A constant voltage of 30 kV (normal polarity) was applied. Samples were hydrodynamically injected for 50 s at 50 mbar (47.9 nL as sample volume).

Preconcentration online strategies were employed to improve method sensitivity. An injection solvent of H_2_O:MeOH (95.5:4.5, *v*/*v*) was used. After sample injection, a plug of BGE was injected for 5 s at 50 mbar to improve injection precision. A schematic diagram showing the steps of the SALLE-CZE-MS/MS method is shown in Appendix A.

### 4.6. MS/MS Parameters

Sheath–liquid consisting of a mixture of 30:69.7:0.3 (*v*/*v*/*v*) MeOH:H_2_O:FA was provided at a flow rate of 10 µL/min, with a 1:100 splitter. The mass spectrometer was operated in positive ionization mode (ESI+) under multiple reaction monitoring (MRM) conditions. Capillary and nozzle voltage was set at 2000 V. The other electrospray parameters under optimal conditions were a dry gas flow rate of 11 L/min with a dry gas temperature of 150 °C; nebulizer pressure of 12 psi; sheath gas temperature of 195 °C, and sheath gas flow rate of 5 L/min. MS/MS experiments were performed by fragmentation of the protonated or di-protonoted molecular ([M+H]^+^ and [M+2H]^2+^), which were selected as the precursor ions in all cases except for MCLY, where the ammonium adduct ([M+H+NH_4_]^2+^) was selected instead. An MRM scan with three time-segments was employed, from 0.10 to 5.40 min transitions corresponding to MCRR and AP B; from 5.40 to 5.95 min MCWR, [D-Asp3]MCLR, MCHilR, MCHtyR, MCLR, MCYR, and NOD; from 5.95 to 7.50 min, MCLW, MCLA, MCLF, AP A, and MCLY were monitored. Collision energies (V) were set between 5 and 75, and product ions were analyzed in the range of 58.0–925.3 *m*/*z*. Optimized MS/MS transitions parameters are summarized in Appendix A.

## Figures and Tables

**Figure 1 toxins-17-00063-f001:**
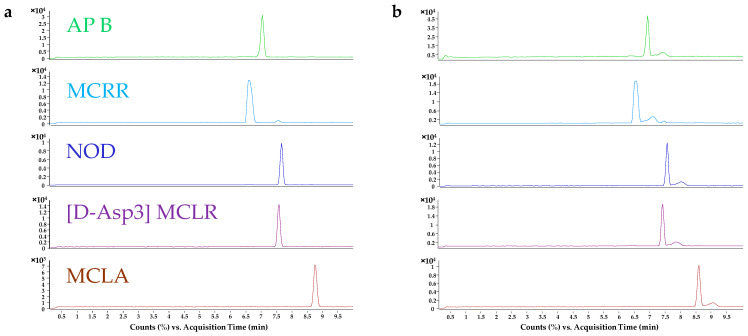
(**a**) Extracted ion electropherograms of different analytes using 50 mM ammonium acetate as BGE. (**b**) Extracted ion electropherograms of different analytes using 40 mM ammonium acetate as BGE. Analyte names are indicated at the left side of the electropherograms and grouped by colors. Analytes not shown in the figure did not have a significant variance in peak shape. Analyte concentrations are: 0.5 µg/mL for AP B, [D-Asp3] MCLR and MCRR; 1 µg/mL for MCLA and NOD.

**Figure 2 toxins-17-00063-f002:**
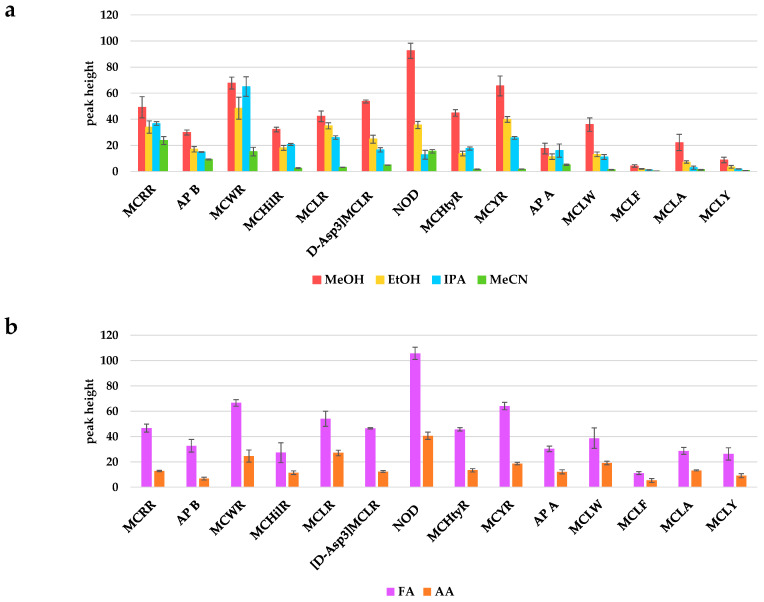
Study of the sheath–liquid composition. (**a**) Organic solvent optimization. Organic solvent: H_2_O:FA (50:49.9:0.1) were the conditions of the experiment. (**b**) Acid nature optimization. MeOH:H_2_O:acid (50:49.9:0.1) were the conditions of the experiment. Error bars represent the standard error (*n* = 9).

**Figure 3 toxins-17-00063-f003:**
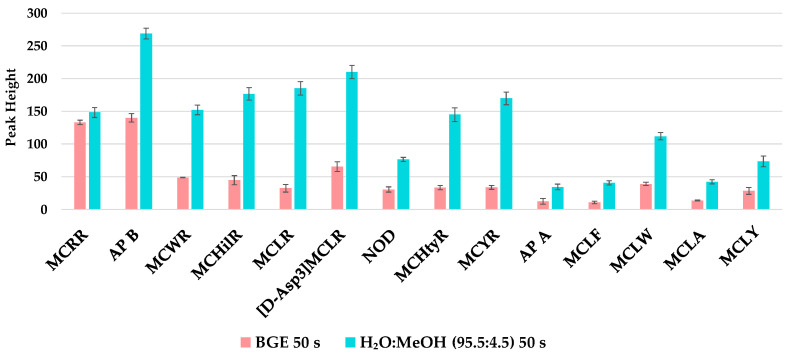
Comparison of the analyte peak height using BGE and H_2_O:MeOH (95.5:4.5, *v*/*v*) as a sample solvent and the same injection time. Error bars represent standard error (*n* = 9).

**Figure 4 toxins-17-00063-f004:**
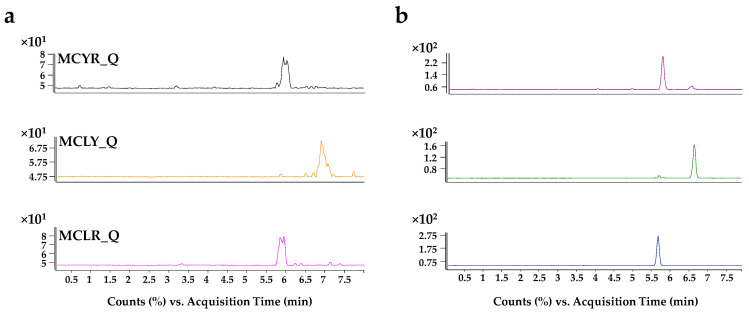
(**a**) Extraction ion electropherograms of the quantification ion of MCYR, MCLY, and MCLR without stacking effect. (**b**) Extraction ion electropherograms of the quantification ion of MCYR, MCLY, and MCLR with stacking effect.

**Figure 5 toxins-17-00063-f005:**
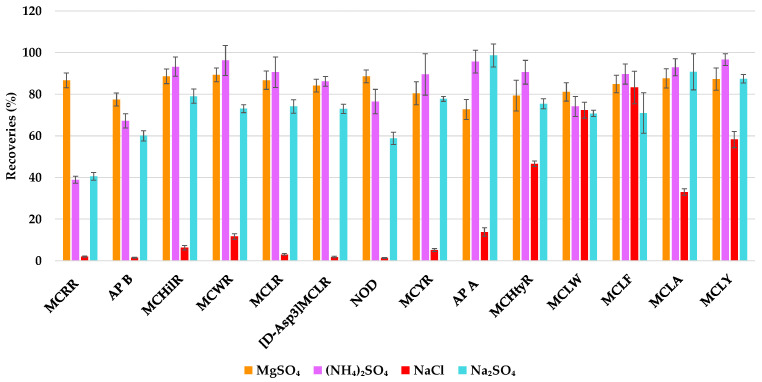
Study of the effect of salt nature in the SALLE procedure on the recoveries of the analytes: MgSO_4_ (orange), (NH_4_)_2_SO_4_ (purple), NaCl (red), and Na_2_SO_4_ (blue). Error bars represent standard error (*n* = 9).

**Figure 6 toxins-17-00063-f006:**
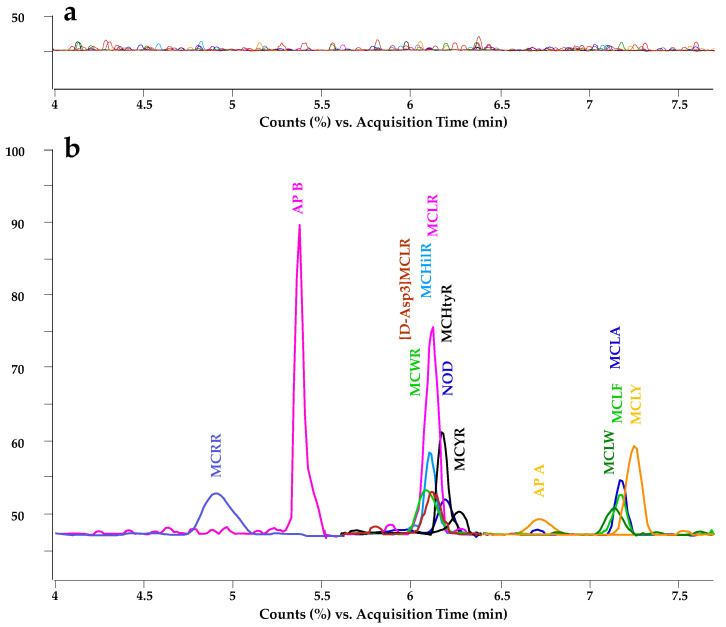
Electropherograms using the proposed SALLE-CZE-MS/MS method for a thermal water sample from Santa Fe (Granada). (**a**) Blank sample. (**b**) Sample spiked before sample treatment. Concentrations: [D-Asp3] MCLR and MCYR 0.04 µg/L; AP B, MCHilR, MCRR, MCHtyR, MCLW 0.08 µg/L; MCLR, MCWR, MCLA 0.12 µg/L; MCLY, NOD, AP A, and MCLF 0.20 µg/L.

**Figure 7 toxins-17-00063-f007:**
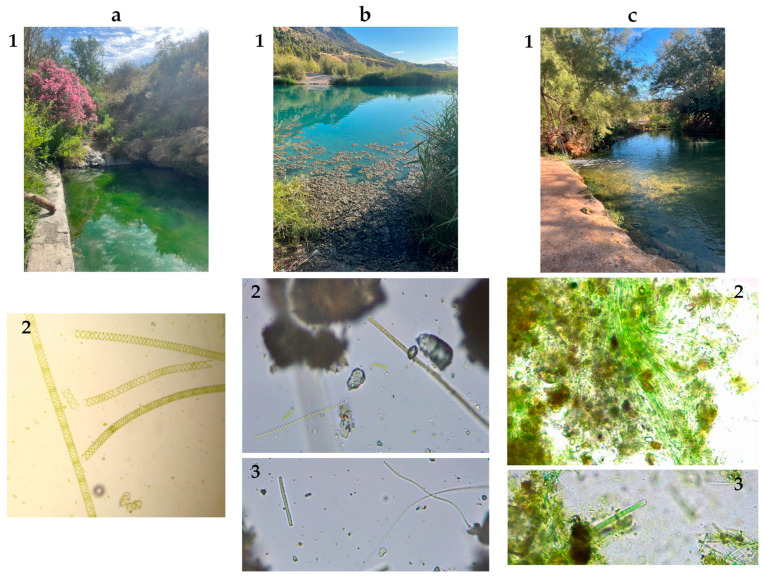
(**a**) (1) Dúrcal thermal spot. (2) Abundant presence of filamentous green microalgae (chlorophycean eukaryote) genus *Spyrogyra*; (**b**) (1) Zújar thermal spot. (2) Micrograph of the genera *Spirulina* (left) and *Phormidium* (right) 40×. (3) Micrograph of the genus *Phormidium* (shorter and thicker on the left) and two finer filaments of the genus *Leptolyngbya* (on the right); (**c**) (1) Santa Fe thermal spot. (2) Abundant filaments of the genus *Leptolyngbya* 40×. (3) Micrograph of abundant *Leptolyngbya* filaments, and a thicker filament (middle-left) of the genus *Phormidium* 40×.

**Table 1 toxins-17-00063-t001:** Performance characteristics of the proposed method.

Analytes	Linear Range (µg/L)	LOQ (µg/L)	LOD (µg/L)	R^2^
MCYR	0.02–0.32	0.02	0.01	0.995
[D-Asp3]MCLR	0.03–0.32	0.03	0.01	0.996
MCHtyR	0.04–0.64	0.04	0.01	0.992
AP B	0.05–0.64	0.05	0.01	0.994
MCHilR	0.05–0.64	0.05	0.01	0.993
MCLW	0.05–0.64	0.05	0.01	0.993
MCRR	0.06–0.64	0.06	0.02	0.994
MCWR	0.05–0.96	0.05	0.02	0.993
MCLR	0.06–0.96	0.06	0.02	0.995
MCLA	0.07–0.96	0.07	0.02	0.992
MCLY	0.09–1.60	0.09	0.03	0.995
NOD	0.09–1.60	0.09	0.03	0.995
AP A	0.11–1.60	0.11	0.03	0.993
MCLF	0.12–1.60	0.12	0.04	0.992

**Table 2 toxins-17-00063-t002:** Matrix effect, recoveries, and precision in the determination of toxic cyanopeptides by SALLE-CZE-MS/MS in a thermal spring water sample from Santa Fe (Granada).

Analytes	Recoveries (%)				Matrix Effect (%)				Intra-day Precision (RSD, %)		Inter-day Precision (RSD, %)	
	L1 *	RSD (%) (L1)	L2 *	RSD (%) (L2)	L1 *	RSD (%) (L1)	L2 *	RSD (%) (L2)	*n* = 9		*n* = 15	
									L1 *	L2 *	L1 *	L2 *
MCRR	85.7	8.3	88.4	8.9	−10.8	9.9	4.6	7.6	8.3	8.3	11.7	14.1
AP B	91.3	12.2	94.6	12.6	−11.7	5.9	−7.5	10.5	11.0	10.0	10.6	14.9
MCWR	95.2	10.6	94.7	11.3	−7.5	9.8	−8.2	7.3	9.4	9.8	13.7	12.5
MCLR	97.2	8.7	98.6	8.4	−8.2	3.2	−6.1	7.4	7.3	6.8	6.5	13.7
MCHilR	98.4	11.8	89.5	9.1	8.3	10.1	7.0	9.0	9.4	7.9	11.8	11.9
[D-Asp3] MCLR	96.4	9.7	95.9	3.6	0.6	5.4	−7.2	6.8	9.7	5.5	9.6	14.4
NOD	99.0	8.8	86.3	8.5	−6.8	7.8	2.9	4.3	10.4	8.1	12.7	6.6
MCHtyR	90.0	8.0	93.9	10.6	−8.4	6.3	−3.6	8.8	9.3	9.6	12.3	7.1
MCYR	100.6	11.8	87.6	8.0	−10.9	10.0	−8.9	10.4	13.0	14.0	13.7	7.5
AP A	97.0	8.3	96.0	12.0	−3.3	5.5	−4.3	11.0	9.8	10.1	14.2	14.6
MCLW	86.7	14.2	94.8	4.6	−7.8	11.9	−3.8	5.7	11.8	9.4	10.0	15.1
MCLF	99.8	12.2	93.0	3.3	−2.6	12.4	−2.2	2.9	13.1	6.0	12.9	15.1
MCLA	95.4	11.0	101.7	9.1	2.7	11.5	−0.1	9.6	10.5	9.2	14.8	14.9
MCLY	93.6	9.4	94.1	7.3	−6.7	7.7	−5.0	4.5	5.8	5.9	9.0	9.1

* Concentration levels L1 and L2 are, respectively: [D-Asp3] MCLR and MCYR 0.04 and 0.16 µg/L; AP B, MCHilR, MCRR, MCHtyR, MCLW 0.04, and 0.32 µg/L; MCLR, MCWR, MCLA 0.12, and 0.48 µg/L; MCLY, NOD, AP A, and MCLF 0.02 and 0.80 µg/L.

**Table 3 toxins-17-00063-t003:** Main aspects of the methods to determine cyanotoxins together with their corresponding AGREE and BAGI Scores.

Sample Prep.	CTXs	Sample (mL)	Extraction Solvent (mL)	TAT (min)	Separation/Detection Technique	AGREE Score	BAGI Score	Ref.
SALLE	9	Salinewater (4)	MeCN (2)	12	UHPLC-MS/MS	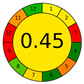	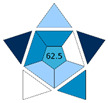	[38]
SALLE	14	Thermal water (7.5)	MeCN (2)	7	CZE-MS/MS	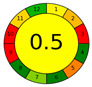	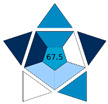	This work

TAT: Total Analysis Time.

## Data Availability

The original contributions presented in this study are included in the article/Appendix A. Further inquiries can be directed to the corresponding author.

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
