# Peer review of "Toxic Cyanopeptides Monitoring in Thermal Spring Water by Capillary Electrophoresis Tandem Mass Spectrometry"

_toxins, 2025, doi:10.3390/toxins17020063_

Round 1
Reviewer 1 Report
Comments and Suggestions for Authors
1. Check the writing errors: there should be no space between numbers and percentage signs (%).
2. Although the authors collected water samples from six natural hot springs in the province of Granada, the sample sizes were still relatively small. That might not be fully representative of the geothermal water conditions in the region. Specifically, the sampling was conducted at only two times (June and September). Why do you choose these two times? That may not be able to capture the seasonal changes of cyanobacterial toxin levels.
3. It is recommended to use principal component analysis (PCA) or other multivariate analysis methods to directly show the data of different sampling points or time points in order to more intuitively observe the correlations and differences between the data.
4. While the method reported in the paper performs well in terms of limit of quantitation (LOQ), lower than many methods in published papers, LOQ for some toxins (such as APA and APB) is still relatively high. This may limit the sensitivity of the method in practical applications. Further optimization of experimental conditions, such as adjusting sample handling steps or instrument parameters, is recommended to reduce LOQ and increase sensitivity.
5. Despite the use of highly efficient methods, the target blue peptide was not detected and the explanations provided were too simple, mainly due to the possibility of inappropriate sampling time and location or toxin concentrations below detection limits. However, there were no further exploration of underlying environmental factors, cyanobacteria growth cycles, or toxins releasing mechanisms, which made the interpretation of the results less than thorough and comprehensive. In contrast, the reasons for undetected toxins are analyzed and discussed in more detail in reference [47].
6. The discussion is not deep enough. The latest research progress and references in related fields should be added to support the views and conclusions of the research. In addition to describing the experimental results, the mechanism of different conditions affecting the separation efficiency and detection sensitivity should be discussed. Combined with the current situation and trend of cyanobacteria toxins pollution, the practical application prospect and challenge of this method in the field of environmental monitoring and public health were discussed.
Author Response
- Check the writing errors: there should be no space between numbers and percentage signs (%).
Done.
- Although the authors collected water samples from six natural hot springs in the province of Granada, the sample sizes were still relatively small. That might not be fully representative of the geothermal water conditions in the region. Specifically, the sampling was conducted only two times (June and September). Why do you choose these two times? That may not be able to capture the seasonal changes of cyanobacterial toxin levels.
The aim of our work is to develop a reliable analytical tool capable of monitoring the presence of cyanotoxin in thermal water. This paper is by no means a study of occurrence. Nevertheless, we have covered as many geothermal spots as possible in this region. In fact, only three points in the province of Granada were left out of our sampling because they have restricted access as they belong to private companies. We believe that six different types of water samples is a reasonable number to demonstrate the reliability of the proposed analytical tool.
Cyanobacteria cycle is yet unknown in hot springs where the ambient temperature is practically maintained throughout the seasons. However, this cycle is quite well known in fresh waters, and we thought it was a good starting point. In this sense, we have chosen these two times of the year to make them correspond with the before and the after of cell rupture. Furthermore, in reference 47, a work focused on thermal crisis and the relation with microcystins, they sampled only once in September, coinciding with our second sampling.
- It is recommended to use principal component analysis (PCA) or other multivariate analysis methods to directly show the data of different sampling points or time points in order to more intuitively observe the correlations and differences between the data.
The data that we would like to observe correlations or differences is the cyanotoxin content. In this sense, PCA would have been a valuable tool for predicting the cyanotoxin content of samples based on a certain combination of water characteristics. Unfortunately, in our case no differences in the cyanotoxin content were found in the data between time points or sampling points, making grouping samples into clusters useless.
- While the method reported in the paper performs well in terms of limit of quantitation (LOQ), lower than many methods in published papers, LOQ for some toxins (such as APA and APB) is still relatively high. This may limit the sensitivity of the method in practical applications. Further optimization of experimental conditions, such as adjusting sample handling steps or instrument parameters, is recommended to reduce LOQ and increase sensitivity.
We are aware that sensitivity is key when determining cyanotoxins. For this reason, sample treatment and instrument parameters were optimized trying to obtain the lowest limits of detection, together with sufficient robustness. For example, a stacking approach has been used for electrophoretic injection. This approach is used in CE only to improve method sensitivity compared to a regular hydrodynamic injection. In fact, the results obtained are the lowest limits published to date in CE-MS for these analytes, to our knowledge. We believe that they may be satisfactory for monitoring thermal hot spring. For example, APA has a LOQ of 0.11 µg/L, that is one order of magnitude below MCLR legal limit (1 µg/L).
- Despite the use of highly efficient methods, the target blue peptide was not detected and the explanations provided were too simple, mainly due to the possibility of inappropriate sampling time and location or toxin concentrations below detection limits. However, there were no further exploration of underlying environmental factors, cyanobacteria growth cycles, or toxins releasing mechanisms, which made the interpretation of the results less than thorough and comprehensive. In contrast, the reasons for undetected toxins are analyzed and discussed in more detail in reference [47].
Again, our aim is to develop a reliable analytical method that can be used to monitor the presence of cyanotoxins in hot springs. It is not our intention to provide explanations beyond the fact that if cyanotoxins are not detected, it is not due to an inadequate analytical method. Hopefully, biologists will find this tool suitable for further studies on the growth cycle of cyanobacteria in hot springs.
After carefully reading reference 47 again, we could not find any explanation for undetected toxins. In fact, they do not analyze water samples but algal samples. Are you sure you mean reference 47 (https://doi.org/10.1016/j.hal.2022.102240)?
- The discussion is not deep enough. The latest research progress and references in related fields should be added to support the views and conclusions of the research. In addition to describing the experimental results, the mechanism of different conditions affecting the separation efficiency and detection sensitivity should be discussed. Combined with the current situation and trend of cyanobacteria toxins pollution, the practical application prospect and challenge of this method in the field of environmental monitoring and public health were discussed.
Two additional papers have been added following the reviewer’s comment.
https://doi.org/10.1002/elps.201900042
https://doi.org/10.1016/j.talanta.2010.05.045
Conditions affecting separation efficiency and detection sensitivity have been extensively studied. Every parameter affecting detection sensitivity was studied with a wide range of values: sheath liquid composition, sheath liquid flow rate, sheath gas temperature, sheath gas flow rate, dry gas temperature, dry gas flow, nebulizer pressure, capillary voltage and nozzle voltage. MS/MS product ions for each analyte have been manually selected in order to get the highest signal for each transition. Collision energy and dwell time were also evaluated and set manually for each transition and each analyte. Moreover, a stacking strategy has also been developed to improve sensitivity.
Considering separation efficiency, background electrolyte was meticulously studied: pH, buffer type, addition of organic solvents, etc. Separation conditions were as well evaluated for every possible parameter: capillary length, applied voltage, temperature of the cassette. Nevertheless, we are happy to include an discuss any particular parameters that the reviewer feels we should include due to their relevance.
Reviewer 2 Report
Comments and Suggestions for Authors
The authors have reported the use of CE - TMS for toxin detection, the study is ell conducted the manuscript may be accepted
Minor comments
Figure Y scale is not clear, present the figure in a better way to enhance visibility
Reduce conclusion
Results section lacks a discussion, add a discussion with reference to published work in related area.
Add a schematic diagram showing the steps involved in the CE-TMS based detection of the toxins so that the readers will have a overall perspective of what is going on
Author Response
The authors have reported the use of CE - TMS for toxin detection, the study is ell conducted the manuscript may be accepted
Minor comments
- Figure Y scale is not clear, present the figure in a better way to enhance visibility
Done
- Reduce conclusion
Done
- Results section lacks a discussion, add a discussion with reference to published work in related area.
Two additional papers have been added following the reviewer’s comment.
https://doi.org/10.1002/elps.201900042
https://doi.org/10.1016/j.talanta.2010.05.045
- Add a schematic diagram showing the steps involved in the CE-TMS based detection of the toxins so that the readers will have a overall perspective of what is going on
Done.
Reviewer 3 Report
Comments and Suggestions for Authors
The authors present a new analytical method for the assessment of cyanobacterial toxins using capillary electrophoresis. The major issue that this reviewer has is that there seems to be a large overlap with reference 38 in terms of SALLE and mass spectrometry. Most of the manuscript describes optimization of the method which is better suited to a Ph.D. thesis and there seems to be little discussion in the "results and discussion" section with very few references. For publication, it is suggested that this manuscript is drastically cut down, removing optimization and any duplicated information from reference 38. This would then leave space for discussion of the method and its application and implications. In addition, the following specific points should be addressed:
1. line 32: "and its" should be replaced with "and their"
2. Line 51: This sentence would read better as "serine/threonine protein 1 (PP1) and 2A (PP2A) phosphatases"
3. Line 106: "mayor" should be "major"
4. Figure 1: This figure is missing numbers and legends on the x-axis.
Author Response
- The authors present a new analytical method for the assessment of cyanobacterial toxins using capillary electrophoresis. The major issue that this reviewer has is that there seems to be a large overlap with reference 38 in terms of SALLE and mass spectrometry. Most of the manuscript describes optimization of the method which is better suited to a Ph.D. thesis and there seems to be little discussion in the "results and discussion" section with very few references. For publication, it is suggested that this manuscript is drastically cut down, removing optimization and any duplicated information from reference 38. This would then leave space for discussion of the method and its application and implications. In addition, the following specific points should be addressed:
We understand that might look like there is an overlap between this paper and reference [38] because we both use salting-out liquid-liquid extraction (SALLE); however, this is not the case because a SALLE procedure can produce extracts perfectly compatible with LC that are not suitable at all for CE. The fact that we use different separation techniques is key. For example, a highly conductive extract can be perfectly suitable for LC but completely ruin the CE separation due to current disruptions. In fact, our optimum extraction conditions differ from those of ref [38]. In addition, we must consider that co-extracted compounds might not interfere in LC but might interfere in CE.
Moreover, there are also significant differences in terms of mass spectrometry as the LC-MS interface is quite different from the CE-MS interface. Although both are based on electrospray, CE-MS requires an additional sheath liquid capable of closing the electric circuit. This can explain that in ref. [38] all analytes are monoprotonated while in our work, 12 of the 14 analytes are diprotonated. Besides, the number of analytes is also different.
We have followed the reviewer’s comment and have included two additional references.
https://doi.org/10.1002/elps.201900042
https://doi.org/10.1016/j.talanta.2010.05.045
Although we would love to include and discuss more papers, unfortunately the number of papers dealing with the determination of cyanotoxins by CE is still scarce. That is why we think our paper is relevant for publication.
- line 32: "and its" should be replaced with "and their"
Done.
- Line 51: This sentence would read better as "serine/threonine protein 1 (PP1) and 2A (PP2A) phosphatases"
Done.
- Line 106: "mayor" should be "major"
Done.
- Figure 1: This figure is missing numbers and legends on the x-axis.
Done.
Round 2
Reviewer 1 Report
Comments and Suggestions for Authors
no more comment
Author Response
We are glad to hear that our responses for Reviewer 1 were satisfactory.